# OpenReview forum: "Beyond Unified Directions: Context-adaptive Representation Steering for LLM Safety Alignment"
_ICLR.cc/2026/Conference — ICLR 2026 Conference Withdrawn Submission_

### Official Review · Reviewer_XwJm · 2025-10-19

**Soundness:** 3
**Presentation:** 3
**Contribution:** 2
**Rating:** 4
**Confidence:** 4

**Summary:**

This paper introduces CA-Steer, a context-adaptive representation steering method for improving safety alignment in large language models (LLMs). Unlike prior steering approaches that apply a single global direction to all contexts, CA-Steer computes token-level context-specific steering vectors by retrieving contextually similar safe and unsafe representations from pre-built banks. It also introduces (1) a sample-level padding strategy to handle retrieval imbalance or emptiness and (2) a steering gate that conditionally applies steering only when the prompt is likely to be unsafe.

**Strengths:**

1. The paper identifies a clear limitation in current steering methods — the “one-size-fits-all” assumption — and introduces context-adaptive steering as a principled extension.
2. Experiments are extensive and systematic, covering multiple open-source LLMs and diverse safety/utility benchmarks.
3. Writing is clear, structured, and professional.

**Weaknesses:**

1. Some experimental results are wrong. For example, in Table 1, ALERT benchmark for qwen with $\text{prompt}_\text{hand}$, the author reports the safety performance is 96.16%. However, the test dataset in ALERT has only 1,000 items. How can you get these results? I think the authors should check the results of your paper (Not only what I mentioned).

2. The method is not novel.  The steering method has been widely used. The authors should cite these articles and add a statement of innovation.

**Questions:**

See Weaknesses.

---

> ### Author Response · Authors · 2025-11-17
> **Author Response to Reviewer XwJm**
>
> Thank you for your careful review and helpful suggestions. We hope that the responses can address your concerns and sincerely welcome further discussion.
>
> **Response #1: Potential errors in experimental results**
>
> >1. Some experimental results are wrong. For example, in Table 1, ALERT benchmark for qwen with $\text{prompt}_{\text{hand}}$, the author reports the safety performance is 96.16%. However, the test dataset in ALERT has only 1,000 items. How can you get these results? I think the authors should check the results of your paper (Not only what I mentioned).
>
>
> We sincerely apologize for the typo in Table 1 regarding the safety performance of Qwen with $\text{prompt}_{\text{hand}}$ on ALERT. The correct result is 96.10% (not 96.16%). **This error stemmed from manual input during final table formatting and is unrelated to experimental procedures or data calculation.**
>
> To ensure rigor, we conducted **a comprehensive recheck of all experimental results** (tables, figures, text descriptions), confirming no other discrepancies between reported results and experimental records. The typo has been corrected in the revised version, and we implemented additional checks (e.g., cross-validation between raw data logs and manuscript tables) to prevent similar issues. We deeply appreciate your attention to detail, which strengthens the manuscript’s accuracy.
>
> **Response #2: Novelty of the method**
>
> >2. The method is not novel. The steering method has been widely used. The authors should cite these articles and add a statement of innovation.
>
>
> To explicitly clarify the novelty of CA-Steer, we supplement a systematic comparison with representative works (e.g.,  RepE [1], CAA [2], STA [3], SRE [4]) across core design dimensions:
>
> - **Core Motivation**: Existing methods rely on unified, "one-size-fits-all" steering directions derived from global dataset patterns. In contrast, CA-Steer is motivated by the need for *context-adaptive steering*, tailoring adjustments to the semantic context of individual prompts, which vary significantly in risk catogories and attack methods.
>
> - **Calculation of Steering Vectors**: Prior approaches rely on dataset-level mean representation differences (ignoring contextual variations). CA-Steer adopts *a hybrid retrieval paradigm*: token-level context-similar retrieval + sample-level adaptive padding (addressing imbalanced subsets), grounding vectors in local+global context.
>
> - **Steering Mechanism**: Conventional methods apply unconditional steering to all inputs, leading to redundant computation and utility degradation for benign prompts. CA-Steer implements *conditional activation via a sample-level retrieval-based gate*, which dynamically skips unnecessary steering for low-risk prompts (avoids over-steering in 99.9% of benign cases, e.g., GSM8K).
>
> - **Key Advantage**: Beyond basic safety improvements of existing methods, CA-Steer achieves: *more safety via context adaptability, robust safety-utility balance (no over-steering), and strong OOD generalization*—addressing prior limitations.
>
> | Comparison Dimension                                          | Existing Steering Methods | CA-Steer (Ours)                                                                         |
> |---------------------------------------------------------------|----------------------------------------------------------------------|-----------------------------------------------------------------------------------------|
> | Core Motivation                                                    | Unified steering direction (one-size-fits-all)      | Context-adaptive steering direction                          |
> | Calculation Method                                           | Difference in Dataset-level mean representaions      | Difference based on hybrid retrieval subsets (token-level retrieval + sample-level padding)       |
> | Steering Mechanism                                            | Unconditional steering for all samples | Conditional steering via sample-level retrieval-based gate |
> | Advantage                                                | - Basic safety improvement                     | - More safety improvement via context adaptability        |
> |              |                | - Safety-utility balance        |
> |            |           | - Strong OOD generalization          |
>
>
>
> [1] Zou, Andy, et al. "Representation engineering: A top-down approach to ai transparency." arXiv preprint arXiv:2310.01405 (2023).
>
> [2] Rimsky, Nina, et al. "Steering llama 2 via contrastive activation addition." Proceedings of the 62nd Annual Meeting of the Association for Computational Linguistics (Volume 1: Long Papers). 2024.
>
> [3] Wang, Mengru, et al. "Beyond Prompt Engineering: Robust Behavior Control in LLMs via Steering Target Atoms." arXiv preprint arXiv:2505.20322 (2025).
>
> [4] He, Zeqing, et al. "Towards LLM guardrails via sparse representation steering." arXiv preprint arXiv:2503.16851 (2025).

---

> ### Comment · Area_Chair_TyLj · 2025-11-28
>
> Dear Reviewers,
>
> Thank you for your time and thoughtful feedback on this manuscript.
>
> The authors have now submitted their rebuttal. If you haven’t already, we kindly ask you to review their responses and consider whether your concerns have been adequately addressed.
>
> Best regards,
>
> AC

---

### Official Review · Reviewer_oFFn · 2025-10-30

**Soundness:** 2
**Presentation:** 3
**Contribution:** 2
**Rating:** 4
**Confidence:** 3

**Summary:**

This paper proposes a context-adaptive representation steering framework (CA-Steer) for improving large language model (LLM) safety alignment. While prior methods typically apply a unified, dataset-level steering direction to all contexts, CA-Steer introduces a more fine-grained and dynamic approach. It retrieves token-level and sample-level contextually similar representations to compute adaptive steering vectors during inference, enabling more precise and effective safety control.

**Strengths:**

1. The paper presents a lightweight method for steering LLM behaviour toward safer responses.
2. The analysis showing that safe and unsafe representations form distinct clusters across different contextual settings is insightful.

**Weaknesses:**

1. Limited generalizability. Since the steering vectors in CA-Steer are context-dependent, the method appears potentially data-intensive, requiring diverse and comprehensive representation banks that cover various contextual settings and risk types.
- Could the authors discuss how CA-Steer scales when contextual coverage is limited, for instance, if certain risk categories or prompt styles are underrepresented in the safety dataset?
2. Confusing design on computing the steering vector. The method computes a steering vector by retrieving contextually similar safe and unsafe representations for the same target token representation $h$.
- How can $h$ be simultaneously similar to both safe and unsafe examples? Representation steering typically assumes that safe and unsafe representations are linearly separable, implying that a token should be close to either safe or unsafe examples but not both.
- The authors report that, on ALERT, 55.1% of retrieved subsets are dominated by safe samples and 22.5% by unsafe ones. Does this suggest that most retrievals are inherently imbalanced and that balanced retrievals are relatively rare? If so, does this mean that computing a steering vector based on both safe and unsafe similar representations may not always be feasible or meaningful in practice?
3. Concerns on the effectiveness of the padding strategy. In Section 4.2, the authors report that Global Padding outperforms Only Padding, even though Global Padding uses dataset-level mean representations while Only Padding uses context-specific samples.
- Could the authors clarify why the global representation performs better than the local, context-dependent one?

**Questions:**

Please refer to the weaknesses.

---

> ### Author Response · Authors · 2025-11-17
> **Author Response to Reviewer oFFn (Part 1/3)**
>
> **Response #1: Clarification on limited generalizability**
>
> >1. Limited generalizability. Since the steering vectors in CA-Steer are context-dependent, the method appears potentially data-intensive, requiring diverse and comprehensive representation banks that cover various contextual settings and risk types.
> >- Could the authors discuss how CA-Steer scales when contextual coverage is limited, for instance, if certain risk categories or prompt styles are underrepresented in the safety dataset?
>
> Due to its reliance on representation bank retrieval, CA-Steer's generalization ability appears limited. However, in practice, **CA-Steer demonstrates stronger generalization ability compared to other steering baselines,** as validated by both existing and supplementary experiments:
>
> 1\) **Existing OOD generalization evidence**: The representation bank is built using ALERT data, which does not explicitly cover all risk categories in our out-of-distribution (OOD) benchmarks (e.g., "ethics and morality" in SafeEdit, "Misinformation" in WildGuard). *Despite this coverage gap, CA-Steer outperforms baselines significantly on these OOD datasets*: compared to the strongest baseline CAA, it improves safety scores by 14.74% on SafeEdit (from 81.78% to 96.52%) and 2.83% on WildGuard (from 93.21% to 96.04%). This suggests CA-Steer’s context-adaptive retrieval can leverage partial contextual patterns to generalize to unseen risk categories.
>
> 2\) **Supplementary experiments on limited coverage scenarios**: To directly address scalability under incomplete contextual coverage, we designed a controlled experiment: we removed all training data except for the "Hate Speech" category, then incrementally added training data for "Criminal Planning" (denoted as "X"), monitoring performance on three risk categories: "X" (Criminal Planning, directly trained); "Y" (Illegal Weapons, semantically close to "X"); "Z" (Sexual Content, semantically distant from "X").
>
> As shown in the Table below, CA-Steer exhibits robust scalability with increasing training data for "X": For "X" itself, safety performance improves by +2.72% (from 96.24% to 98.96%); For "Y" (close to "X"), it improves by +1.91% (from 96.32% to 98.23%); Even for "Z" (distant from "X"), it still improves by +1.62% (from 96.29% to 97.91%). In contrast, CAA (a global steering baseline) shows limited generalization: it improves by only +1.83% for "X", +0.55% for "Y", and even a slight drop (-0.01%) for "Z". This confirms that CA-Steer’s hybrid retrieval (token-level + sample-level) enables it to transfer contextual patterns across other categories well, even when coverage is limited.
>
> *We believe that CA-Steer's strong generalization is because, for attack samples with unknown patterns, using the closest known pattern as a reference to calculate the steering direction is already a locally optimal solution.*
>
> | Method            | #Training Data of Category "X" | Category "X" | Category "Y" (close to "X") | Category "Z" (far from "X") |
> |-------------------|---------------------------------|:--------------:|:-----------------------------:|:-----------------------------:|
> | Vanilla           | -                               |    94.82     |            93.87            |            94.24            |
> | CAA               | 0%                              |    96.06     |            95.27            |         97.66            |
> |                | 20%                             |    96.58     |            95.89            |         98.05            |
> |                | 50%                             |    97.14     |            95.94            |            97.35            |
> |                | 100%                            |    97.89     |            95.82            |            97.65            |
> | **CA-Steer (Ours)** | 0%                              |   96.24  |        96.32            |            96.29            |
> | | 20%                             |   97.68  |        97.14            |            97.46            |
> |  | 50%                             |   98.54  |        97.75            |         **97.95**            |
> | | 100%                            |   **98.96**  |        **98.23**            |         97.91            |

---

> ### Author Response · Authors · 2025-11-17
> **Author Response to Reviewer oFFn (Part 2/3)**
>
> **Response #2: Steering vector computation**
>
> >2. Confusing design on computing the steering vector. The method computes a steering vector by retrieving contextually similar safe and unsafe representations for the same target token representation $h$.
> >- How can $h$ be simultaneously similar to both safe and unsafe examples? Representation steering typically assumes that safe and unsafe representations are linearly separable, implying that a token should be close to either safe or unsafe examples but not both.
> >- The authors report that, on ALERT, 55.1% of retrieved subsets are dominated by safe samples and 22.5% by unsafe ones. Does this suggest that most retrievals are inherently imbalanced and that balanced retrievals are relatively rare? If so, does this mean that computing a steering vector based on both safe and unsafe similar representations may not always be feasible or meaningful in practice?
>
>
> We apologize for any confusion caused by "representation $h$", and offer the following clarification.
>
> 1\) **Clarification regarding "representation $h$": encoding complex contextual information, not binary safety attributes.**
>
> In our work, "safe representations" and "unsafe representations" refer to the contextual representations extracted from "query + safe response" and "query + unsafe response", respectively, rather than representations that only encode binary safety attributes. These representations themselves encode rich latent variables beyond binary safety, such as risk categories, attack methods, and dialogue domains. **While safe/unsafe representations are linearly separable in the safety-specific subspace, they can overlap in broader semantic subspaces** (e.g., two prompts about "weapon use" may share linguistic structures but differ in safety intent). Thus, $h$ can be semantically similar to both safe and unsafe samples via non-safety subspaces, making retrieval of both sets meaningful for context-adaptive steering.
>
> 2\) **Concerns about "imbalanced retrieval":**
>
> Since the representation bank is token-level, with millions of entries, a perfectly balanced retrieval subset ($|S^{safe}|=|S^{unsafe}|$) is quite rare. **This imbalance does not mean that steering vector computation based on similar representations is infeasible.** This can be seen in Table 4 of the manuscript, where Global Padding (token-level retrieval + CAA) scores 94.85 on SafeEdit, significantly higher than CAA's 81.78. This indicates that even with an imbalanced retrieval subset, token-level retrieval can still provide auxiliary contextual adaptive information for a unified steering direction (CAA), thus achieving more accurate safe steering. We believe that **the effectiveness of the imbalanced retrieval subset still exists because CA-Steer is equivalent to performing test-sample specific calibrations on $h_{safe}$ and $h_{unsafe}$ within the global steering vector ($h_{safe}-h_{unsafe}$), making the steering vector more accurate. Imbalance only affects the strength of these two calibration.**
>
> In addition, we specifically designed a sample-level context-adaptive padding strategy to solve the problem of imbalanced subsets in token-level retrieval while preserving contextual relevance, thereby further improving model safety.

---

> ### Author Response · Authors · 2025-11-17
> **Author Response to Reviewer oFFn (Part 3/3)**
>
> **Response #3: Effectiveness of the padding strategy**
>
> >3. Concerns on the effectiveness of the padding strategy. In Section 4.2, the authors report that Global Padding outperforms Only Padding, even though Global Padding uses dataset-level mean representations while Only Padding uses context-specific samples.
> >- Could the authors clarify why the global representation performs better than the local, context-dependent one?
>
> We apologize for any confusion caused by "padding strategy experiment settings", and we clarify the padding strategies and provide more valid comparisons below:
>
> **1\) Clarification of key padding strategy definitions**
>
> - *Global Padding*: Token-level retrieval subsets are padded with CAA’s dataset-level global vectors (not using global vectors alone). The final steering vector is calculated as the difference in means of token-level retrieval subsets and global padding.
> - *Only Padding*: Steering vectors are computed solely using sample-level retrieval mean representations (no token-level retrieval).
>
> These two strategies are not directly comparable, as they differ not only in a single setting. We provide two more meaningful comparisons to validate the value of context-adaptive padding as following.
>
> **2\) Valid comparisons for padding strategy effectiveness**
>
> - *CAA vs. Only Padding*: Directly compares global vs. sample-level context-adaptive vectors (no token-level retrieval). *Only Padding* achieves 86.37 on SafeEdit, outperforming CAA (81.78)—verifying that **sample-level context-adaptive vectors are more accurate than global uniform vectors**.
>
> - *Global Padding vs. CA-Padding*: Compares global vs. sample-level padding for token-level retrieval subsets. CA-Padding (sample-level context-adaptive padding) achieves 96.52 on SafeEdit, outperforming Global Padding (94.85)—confirming that **context-adaptive padding further enhances the utility of token-level retrieval, outperforming global uniform padding**.
>
> |     Strategy                                                  |     ALERT     |     SafeEdit    |
> |---------------------------------------------------------------|:---------------:|:-----------------:|
> |     Vanilla LLM                                               |     96.70     |     70.44       |
> |     CAA (global vector)                                        |     97.40     |     81.78       |
> |     No Padding (token-level retrival)                          |     96.80     |     81.97       |
> |     Only Padding (sample-level retrival)                       |     97.60     |     86.37       |
> |     Global Padding (token-level retrival + CAA padding)          |     98.20     |     94.85       |
> |     CA-Padding (token-level retrival + sample-level padding)    |     **98.70**     |     **96.52**       |

---

> ### Comment · Area_Chair_TyLj · 2025-11-28
> **Rebuttal Review Request**
>
> Dear Reviewers,
>
> Thank you for your time and thoughtful feedback on this manuscript.
>
> The authors have now submitted their rebuttal. If you haven’t already, we kindly ask you to review their responses and consider whether your concerns have been adequately addressed.
>
> Best regards,
>
> AC

---

### Official Review · Reviewer_nZKu · 2025-11-02

**Soundness:** 3
**Presentation:** 3
**Contribution:** 2
**Rating:** 4
**Confidence:** 4

**Summary:**

The paper presents a method for LLM representation steering for inference-time safety. The key novelty behind the presented method is incorporating context information from the input prompt and partial response to compute the steering vector for each token. The paper also presents a simple gating mechanism that decides whether a token needs steering or not. Evaluation on safety and utility benchmarks shows improvements to the model safety and a minor degradation to the model utility (which is clearly demonstrated to be due to the introduced gating mechanism).

**Strengths:**

1. The presented method is simple and intuitive and is shown to introduce a relatively small compute overhead.

2. The evaluation clearly demonstrates the effectiveness of the method compared to previous work.

**Weaknesses:**

1. The safety evaluation is somehow limited: it is based on 3 benchmarks with all 3 models evaluated already are doing reasonably well on 2 of them. It would be interesting to demonstrate the effectiveness of the method against adversarial jailbreak attacks as well. That would be a stronger evidence on the robustness of the presented approach.

**Questions:**

Would you please elaborate on the training data? its size, source, structure, etc.

---

> ### Author Response · Authors · 2025-11-17
> **Author Response to Reviewer nZKu**
>
> We thank the reviewer for the constructive comments and helpful suggestions, which have helped strengthen the robustness validation of our method and clarify key details of our work. Below are our detailed responses, and we sincerely welcome further discussion.
>
>
> **Response #1: More safety evaluation about adversarial jailbreak attacks**
>
> >The safety evaluation is somehow limited: it is based on 3 benchmarks with all 3 models evaluated already are doing reasonably well on 2 of them. It would be interesting to demonstrate the effectiveness of the method against adversarial jailbreak attacks as well. That would be a stronger evidence on the robustness of the presented approach.
>
> To address the concern about limited safety evaluation, we have supplemented experiments on two adversarial jailbreak benchmarks: JailbreakBench (300 samples) and WildJailbreak (2000 samples), alongside our original evaluations on ALERT, SafeEdit, and WildGuard.
>
> As shown in the table below, **CA-Steer maintains its superiority on both adversarial datasets**: it achieves 87.33% on JailbreakBench and 88.90% on WildJailbreak, outperforming the best baseline CAA by 4.00% and 4.20% respectively. This consistent advantage across both standard and adversarial benchmarks confirms that CA-Steer’s context-adaptive design not only enhances general safety but also strengthens resistance to targeted jailbreak attacks.
>
> These additional experiments will be included in the revised manuscript to better demonstrate the method’s robustness and generalization.
>
> | Method               | ALERT  | SafeEdit | WildGuard | JailBreakBench | WildJailBreak | AVG   |
> |----------------------|:--------:|:----------:|:-----------:|:----------------:|:---------------:|:-------:|
> | Vanilla              | 96.70  | 70.44    | 90.25     | 81.67          | 80.20         | 83.85 |
> | RepE                 | 97.20  | 75.55    | 90.49     | 84.33          | 81.95         | 85.90 |
> | CAA                  | 97.40  | 81.78    | 93.21     | 83.33          | 84.70         | 88.08 |
> | STA                  | 97.50  | 81.11    | 91.10     | 82.67          | 83.00         | 87.08 |
> | SADI                 | 97.40  | 74.31    | 93.57     | 81.33          | 83.90         | 86.10 |
> | **CA-Steer (ours)**      | **98.70** | **96.52** | **96.04** | **87.33**       | **88.90**      | **93.50** |
>
> **Response #2: Elaboration on training data**
>
> >Would you please elaborate on the training data? its size, source, structure, etc.
>
> We appreciate the reviewer’s inquiry about the training data details. The core dataset used for constructing the representation bank is elaborated as follows (with full specifications in Appendix B.1.1 of the paper), focusing on its size, source, and structure:
>
> **Source**: The primary data comes from **the adversarial DPO subset of ALERT**, a widely used dataset for safety alignment. This subset is specifically designed to include adversarial prompts and their corresponding safe/harmful responses, making it suitable for learning contrastive safety representations.
>
> **Structure**: Each raw sample in the ALERT subset follows **a triple structure**:
> An adversarial prompt (e.g., subtly crafted queries attempting to elicit harmful outputs);
> A "chosen" safe response (aligned with safety guidelines);
> A "rejected" harmful response (violating safety norms).
>
> **Size**: The raw subset contains 31,000 such triples. To ensure the quality of contrastive representations (critical for distinguishing safe vs. harmful patterns), we filtered out 5,662 samples where the "rejected" response lacked genuine harmful content (e.g., false positives or ambiguous violations). This resulted in **25,338 high-quality prompt-response triples for the representation bank**. The prompts in the unfiltered samples (5,662) were held out as an in-distribution test set, ensuring the bank’s training data and evaluation data are strictly separated.

---

> ### Comment · Area_Chair_TyLj · 2025-11-28
> **Rebuttal Review Request**
>
> Dear Reviewers,
>
> Thank you for your time and thoughtful feedback on this manuscript.
>
> The authors have now submitted their rebuttal. If you haven’t already, we kindly ask you to review their responses and consider whether your concerns have been adequately addressed.
>
> Best regards,
>
> AC

---

### Author Response · Authors · 2025-11-18
**Author Response to All Reviewers**

We thank all the reviewers for their insightful comments and helpful suggestions. We hope our responses and paper updates alleviate the concerns raised.

Following the reviewers’ feedback, we have updated our manuscript mainly with the following contexts:

- **Evaluation on adversarial jailbreak attacks** (Section 3.3-Line290): Following the suggestion from **Reviewer nZKu**, we supplemented experiments on adversarial jailbreak benchmarks to further enhance safety evaluation, and the consistent improvements also enhance the robustness of the method.
- **Scalability under incomplete contextual coverage** (Section 3.4-Line307): Following the suggestion from **Reviewer oFFn**, we added a controlled experiment to explore CA-Steer's scalability under incomplete contextual coverage and validate its strong generalization.
- **Typo correction and rechecking** (Table 1-Line231): Following the feedback from **Reviewer XwJm**, we corrected the manual typo (96.10 instead of 96.16), and implemented additional checks to prevent similar issues, which strengthens the manuscript’s accuracy.

We have highlighted the corresponding modifications in the manuscript with color blue for your convenience. Again, thank you for your hard work. We believe your input has already helped improve the paper and look forward to further engaging with you during the discussion. Please see our replies to each of you below.

---

### Note · Authors · 2025-12-02

I have read and agree with the venue's withdrawal policy on behalf of myself and my co-authors.